# Competitive intelligence and its impact on innovations in tourism industry of China: An empirical research

Yanli Bao [ID] *

Business School of Hohai University, Hohai University, Nanjing, People's Republic of China

* sarah1826@126.com

## Abstract

Competitive intelligence (CI) has attracted much attention in innovation research, but most of existing literature studies CI in technological innovations in manufacturing industry, with little empirical research in context of service businesses. This paper first analyzes CI of service businesses and then uses covariance-based structural equation modeling (SEM) on a data of 333 got from the survey in tourism enterprises of east China to test the effect of customer CI, opponent CI, and supplier CI on service innovations in China's service industry. Results show that opponent CI and supplier CI have positive influence on both exploratory and exploitative service innovation. Customer CI has more obvious positive influence on exploratory service innovation than on exploitative service innovation.

**Data Availability Statement:** All relevant data are within the paper and its Supporting Information files.

## 1 Introduction

A growing consensus recognizes that service firms have been found to rely heavily on information flows within and outside their boundaries, and non-R&D innovations which use much external knowledge [1–4]. During the last decade, organizations are changing their innovation manners from a completely in-house and closed way [5] to a more open and collaborative way that involves customers, suppliers, research institutes, and even competitors [6, 7]. Service firms also are likely to collaborate frequently with their customers and suppliers, which play a positive role on firm innovation performance [8, 9]. And service innovation is based on certain knowledge resources, information and technology. Thus, service firms have to carry out proper searching for, understanding and utilizing the latest CI. This practice has created a need for using competitive intelligence(CI) for innovation in service firms.

Previous studies have indicated that the more information about innovation the companies get, the more selections of innovation they will get and the more likely they are to succeed [10]. For innovation, knowledge and information is recognized as being a critical resource [11] and acquiring CI among organizations could effectively promote innovations [12].

Despite these significant contributions, studies that analyze CI and service innovation in service firms are still scarce [13]. Therefore, it is necessary to make novel theoretical and empirical investigations in this respect which will shed new light on the competitive behaviors of service firms. Complementary to most of the existing research, this study investigates the

**Funding:** This research is supported by Philosophy and Social Science Foundation of 2014 China National Education Department.

**Competing interests:** The authors have declared that no competing interests exist.

effects of CI on innovation of service business in considering customer, opponent and supplier CI as well as their mutual relations. We test our hypotheses by means of structure equation modeling using 333 data collected with a survey of tourism enterprises in east China. In the next section, the theoretical background and research hypotheses are illustrated, followed by the research method, data analysis, research results and discussions as the next several parts.

## 2 Theoretical background

### 2.1 Competitive intelligence

Society of Competitive Intelligence Professional (SCIP) is an authoritative body in CI. In 2003, it defined CI as the systematic and ethical collection, analysis and management of external information that can affect the planning, decision-making and business operation. CI has been listed as the fourth reason for the survival of enterprises after capital, technology and talent [14]. Entering the era of knowledge-based economy, the degree of informationization in China is getting higher and higher. Enterprise CI has gradually become one of the decisive factors for the survival and development of enterprises. In general, CI includes information about competitors, customers, suppliers and related technologies [15]. Beal(2000) regards enterprise's customer intelligence, supplier intelligence, opponent intelligence as the competitive environment in which the enterprise lives [16].

### 2.2 Service innovation

Innovativeness of enterprises refers to a "firm's capacity to engage in innovation through the introduction of new processes, products, or ideas" [17]. Service innovation refers to innovation in the field of services, proposing new service concepts, carrying out new service processes, and applying new technologies to improve or change existing services, process and service products [18, 19].

Innovation can be classified into four types, i.e. product innovation, process innovation, organizational innovation and marketing innovation [20]. But according to the scope and knowledge base, innovation is divided into exploratory innovation and exploitative innovation [21, 22]. Exploratory innovation is a large-scale and radical innovation activity with the intention to find out new possibilities. Exploitative innovation is a small-scale and gradual innovation activity with the intention to improve the existing status [23]. Therefore, academia believes that service innovation can also be divided into exploratory service innovation and exploitative service innovation according to the classification of common innovation. Exploratory service innovation refers to serving customers through the use of disruptive or new technologies, new idea or new service process; Exploitative service innovation leverages current capabilities to develop products and services to serve customers better [24].

### 2.3 CI and service innovation

External consultants and knowledge acquisition on opponents, customers and suppliers all have effects on innovation activities and performance [25, 26]. Firms have recognized the importance of information acquired from competitors in the industry, and customer and suppliers [27]. Some scholars found that professional supplier information is the most effective in promoting innovation, and customer information is the most relevant to innovation process [28].

According to our observation on service businesses in China, we find that their innovation is also relying on these three kinds of CI. First, customer CI. Service firms often collect customer CI through customer visits, satisfaction feedback, and evaluation of existing services.

The customer CI then will be used to improve services or explore new services. Second, opponent CI. Service firms get opponent CI about products/services, R&D, customers, marketing, cost, technology, which can help enterprises understand their strengths and weaknesses and analyze the differences between their own services and competitors', and then innovate, imitate or launch innovative services with more advantages. Third, supplier CI. For service businesses, most technological innovations come from suppliers. Therefore, timely access to information on new products, technologies and ideas developed by suppliers can help firms to carry out service innovation as soon as possible.

Therefore, this paper divides CI that influences innovation in service businesses into three elements, namely, customer CI, opponent CI, and supplier CI. Based on the theory of service innovation and features of CI, this paper illustrates how different kinds of CI affect service innovation to reveal the mechanism of CI on service innovation.

**2.3.1 Effect of customer CI on service innovation.**   Customer CI is the source of knowledge and information for service innovation, because products are ultimately customer-oriented. Customer demand promotes enterprises to open up new services or improve existing service technologies or product functions. Customers have the ability to influence innovation [29].

Based on the classification of customer knowledge by Zhang Hongqi and Lu Ruoyu [30], this paper divides customer CI into four main aspects, namely, basic customer CI, customer satisfaction CI, customer demand CI, and CI of customers' participation in innovation. Firstly, basic customer CI and customer satisfaction CI play an active role in service innovation. When enterprises understand customer information, effectively convey customer opinions, and settle customer complaints, the number of service innovation will increase [31]. Secondly, in order to carry out innovations, enterprises must tap the potential needs of customers, collect and analyze their demand, which can help to identify market demand, generate new service concepts and products, and realize exploratory and exploitative service innovation. Thirdly, CI about customer participation in innovation is of positive significance to both exploratory and exploitative service innovation in forms of service interface, service delivery system and service technology. In the process of service innovation, customers can be cooperative designers or producers of new products and services [32].

Therefore, this paper proposes hypotheses:

H1: A higher emphasis on customer CI positively influences exploitative service innovation.

H2: A higher emphasis on customer CI positively influences exploratory service innovation.

**2.3.2 Effect of Opponent CI on service innovation.**   Opponent analysis is the soul of CI [33]. Opponent CI refers to identifying competitors, analyzing their strength, predicting their strategies, especially evaluating their new products and main products in the aspects of price, cost, profit, development and design ability, marketing strategy as well as opponent's strengths, weaknesses and their recognition of customer needs [34].

Opponent CI in service businesses can be classified into three main aspects, i.e. Opponents' daily operation CI, R&D CI and marketing CI of new services. Through opponent CI, enterprises can also grasp their latest developments, especially CI about new products and new services development, which is conducive to enterprises' imitative innovation, which is a major form of exploitative service innovation. Enterprises can optimize their own knowledge resources and improve service by learning from their opponents [35]. They can also realize exploratory service innovation through developing new products and technology with the help of opponent CI.

Based on this, this paper proposes the following hypotheses:

H3: A higher emphasis on opponent CI positively influences exploitative service innovation.

H4: A higher emphasis on opponent CI positively influences exploratory service innovation.

**2.3.3 Effect of supplier CI on service innovation.** Suppliers have a crucial role in improving firms' innovation performance [36, 37]. Supplier involvement into innovation processes has been recognized as a potential source of sustainable competitive advantage, even though the literature is not fully consistent. Supplier is one of the major innovation drivers of Italian service businesses [38]. It is also one of the sources of service innovation and has a positive impact on enterprise innovation activities. In some industries, such as tourism industry and hotel industry, technological innovation is mainly dominated by suppliers [39].

Suppliers affect the types of innovation. In the context of emerging economy, local suppliers' absorptive capacity is critically important in spurring exploitative and exploratory innovation [40]. Suppliers can not only introduce new materials and technologies into existing products and services to meet current needs, but also help enterprises develop new products and services [41]. Suppliers' participation in service innovation can provide innovative ideas, key technologies, raw materials and so on [42]. In addition, suppliers are innovative and their activities such as developing new production methods, adopting new processes, raw materials, new technologies or applying new business models have a positive impact on product and service innovation [43]. Suppliers' contribution assumes various forms, such as supply of innovative components and product/process technologies, or joint product development projects [44]. Moreover, the inventory CI and marketing CI of suppliers have a tremendous impact on enterprises' daily operations. And their price changes and shortages of raw materials, equipment, technology and human resources will also affect whether enterprises can carry out exploratory and exploitative service innovation or not.

Based on this, this paper proposes:

H5: A higher emphasis on supplier CI positively influences exploitative service innovation.

H6: A higher emphasis on supplier CI positively influences exploratory service innovation.

## 3. Methodology

### 3.1. Ethics statement

The study was approved and supported by the institutional review board of Wuxi Tourism Association, Jiangsu, China. The subject of this manuscript is CI and service innovation situation in companies rather than human beings. All people interviewed with the questionnaire provided their consent by answering the questions. Their names and personal information are kept secret. Therefore, they are free to express their feelings about the CI and service innovation in their companies.

### 3.2. Data collections

To test the hypotheses and the model, they had to be converted into a questionnaire. Each construct is represented by a set of indicators which form the questions in the survey. All questions were measured on a positive-to-negative 7-point Likert scale. Questions on the CI and service innovation give a statement and ask for the level of agreement on the following scale: "Strongly agree—predominantly agree—rather agree—neutral—rather disagree—predominantly disagree—strongly disagree." The questionnaire was discussed intensively within our research institute and pre-tested independently with 5 managers from service businesses which were

not included in the sample. These 5 managers all have more than 10 years working experiences in star hotels, travel agencies, tourist attractions, or other service companies. Based on the discussions, the questionnaire was modified.

The firms selected for this study are employees of star hotels and tourism companies of more than 20 staff in China, because tourism industry has the typical characteristics of service industry and a huge amount in China.

Data was collected in two stages. First, in pre-survey, 100 questionnaires were distributed and 94 valid questionnaires were returned. In pre-survey, Cronbach's alpha coefficient and factor load of the scale were calculated by SPSS 23 software, and the item was deleted according to relevant standards. Second, with the refined questionnaire, the investigator gets approval from the administrators of tourism companies and sends an invitation letter out through e-mail to express the need for collection of empirical data concerning service innovation experience in using CI. The administrators then forwarded the message to their staff via email and instructed the receivers to click a hyperlink and redirected them to an online questionnaire system. Consequently, 400 invitation letters were sent to the staff in tourism industry through e-mail. In order to improve the return rate, another follow-up invitation letter was sent to non-responding staff with the same aforementioned procedure after a week. Finally, 362 staff had finished and returned the questionnaire. Altogether 333 valid questionnaires were obtained after deleting unqualified questionnaires, with an effective return rate of 83.25%.

### 3.3. Measures

Measurement items were selected based on a careful literature review. The results from pre-survey showed that there is no bias.

The scale of enterprise CI is modified by the relevant scales used in empirical researches. The scale of customer CI is made by revising Zhang Hongqi and his partners' scale (2013) [45]. Four items were adapted to measure the extend of customer CI(CCI), including basic customer information, customer demand, customer satisfaction and customer participation in innovation. The opponent intelligence (OCI) scale is prepared in 3 aspects, i.e. competitor's daily operation CI, R&D CI and marketing CI of new services. The scale of supplier CI(SCI) is developed on the basis of Gales, Mansour-Cole (1995) [46] and interviews with service business owners. Three are 3 items describing supplier CI including supplier inventory, R&D, and marketing.

The scale of service innovation adopts the scale of exploitative innovation(ETSI) and exploratory innovation(ERSI) developed by Fu Xiao et al. (2012) [47] to assess the extent to which a firm has engaged in innovation activities and has implemented service innovation activities to improve existing service–market positions with 8 items.

## 4. Data analysis

The data analysis of this study was conducted using structural equation modeling (SEM) technique and followed the two-step approach of for assessing the measurement and structural models respectively [48]. SEM is a powerful statistical research technique and it is very flexible in the types of theoretical models to be tested for analyzing the causal relationships between multiple-item constructs [49]. In addition, SPSS and AMOS are adopted as the tools of data analysis.

### 4.1 Instrument validation

An initial exploratory factor analysis is carried out to verify the internal structure of variables. And the confirmatory factor analysis is made on three dimensions of enterprise CI and two

Table 1. Component matrix after rotation of enterprise CI (N = 121).

| | Component | | |
|---|---|---|---|
| | 1 | 2 | 3 |
| CCI1 | -.133 | | |
| CCI2 | .953 | | |
| CCI3 | .932 | | |
| CCI4 | .911 | | |
| OCI1 | | .963 | |
| OCI2 | | .978 | |
| OCI3 | | .943 | |
| SCI1 | | | .940 |
| SCI2 | | | .947 |
| SCI3 | | | .922 |

kinds of service innovation to evaluate their fitness. We selected 121 questionnaires randomly from valid questionnaires for exploratory factor analysis in order to test reliability and validity.

Firstly, exploratory factor analysis is carried out on "Enterprise CI". KMO = 0.742, greater than 0.7, and Bartlett statistic is significantly equal to 0. 000 at 45 degrees of freedom, which is suitable for factor analysis.

Component matrix after rotation of each factor is shown on Table 1. According to the requirement that the characteristic root is greater than 1 and the maximum factor load is greater than 0.5, the factor load of CCI1 (basic customer CI) in the original scale is less than 0.5, which indicates that the measurement information of this item is not accurate enough, so it is removed from the scale.

The results of exploratory factor analysis on the scale which has removed Basic Customer CI(CCI1) shows that each item is distributed in three factors according to expectation, and the factor load has a good distinction among the three factors. Thus, the new revised scale of the "Enterprise CI" has a good validity.

Next, the reliability of each factor is analyzed to test the internal consistency among the items that passed the exploratory factor analysis. The results show that the overall correlation coefficients of all items are greater than 0.8, and the Cronbach Alpha coefficients of all variables are greater than 0.8. Therefore, there is good internal consistency among the items of the variables of "Enterprise CI".

Secondly, exploratory analysis is made on "service innovation". KMO = 0.833. As shown on Table 2, two factors are extracted according to the requirement that the characteristic root should be greater than 1 and the maximum factor load is greater than 0.5.

Table 2. Component matrix after rotation of service innovation (N = 121).

| | Component | |
|---|---|---|
| | 1 | 2 |
| ETSI1 | .779 | |
| ETSI2 | .839 | |
| ETSI3 | .905 | |
| ETSI4 | .826 | |
| ERSI1 | | .786 |
| ERSI2 | | .836 |
| ERSI3 | | .682 |
| ERSI4 | | .801 |

**Table 3. Reliability test of CI and service innovation (N = 212).**

|  | Corrected item and total correlation | Cronbach Alpha of Corrected item | Cronbach Alpha |
|---|---|---|---|
| CCI2 | .824 | .872 | |
| CCI3 | .841 | .858 | .912 |
| CCI4 | .805 | .888 | |
| OCI1 | .807 | .888 | |
| OCI2 | .854 | .849 | .913 |
| OCI3 | .812 | .884 | |
| SCI1 | .823 | .875 | .911 |
| SCI2 | .870 | .833 | |
| SCI3 | .780 | .906 | |
| ERSI1 | .770 | .908 | |
| ERSI2 | .852 | .883 | .918 |
| ERSI3 | .866 | .880 | |
| ERSI4 | .785 | .903 | |
| ETSI1 | .842 | .896 | |
| ETSI2 | .831 | .900 | .924 |
| ETSI3 | .831 | .900 | |
| ETSI4 | .800 | .910 | |

Next, the reliability of each factor is analyzed to test the internal consistency among the items. The results show that the overall correlation coefficient of all items is greater than 0.6, and the Cronbach Alpha coefficient of each variable is greater than 0.7. Therefore, there is good internal consistency among the items of the variables of service innovation.

Thirdly, the confirmatory factor analysis is done using the remaining 212 samples to ensure that the factor structure of all variables tested is consistent with the previous concepts. The reliability of customer CI, opponent CI, supplier CI, exploratory and exploitative service innovations are analyzed. The results are as shown on Table 3 that each variable index meets the reliability index requirements mentioned above and passes the reliability test. Thus, we can be sure that the consistency of variable measures is good.

Then the initial structural equation model is analyzed and calculated by AMOS software. The results of enterprise CI are: CMIN/DF = 2.582, GFI = 0.940, NFI = 0.958, RFI = 0.937, IFI = 0.974, TLI = 0.961, CFI = 0.974, all greater than 0.9, close to 1, RMSEA = 0.087, less than 0.1. The path coefficients are statistically significant at the level of $P < 0.001$(see Table 4). The results of "service innovation" are CMIN/DF = 2.721. GFI = 0.941, NFI = 0.964, RFI = 0.947,

**Table 4. Measurement model fitting results of enterprise competitive intelligence (N = 212).**

|  |  | Estimate | S.E. | C.R. | P |
|---|---|---|---|---|---|
| CCI3 | CCI | 1.000 | | | |
| CCI2 | CCI | 1.137 | .069 | 16.572 | *** |
| CCI1 | CCI | 1.101 | .067 | 16.403 | *** |
| OCI3 | OCI | 1.000 | | | |
| OCI2 | OCI | 1.052 | .058 | 18.086 | *** |
| OCI1 | OCI | .958 | .059 | 16.132 | *** |
| SCI3 | SCI | 1.000 | | | |
| SCI2 | SCI | 1.143 | .069 | 16.454 | *** |
| SCI1 | SCI | 1.233 | .077 | 16.019 | *** |

**Table 5. Measurement model fitting results of service innovation (N = 212).**

|  |  |  | Estimate | S.E. | C.R. | P |
|---|---|---|---|---|---|---|
| ETSI1 | <— | ETSI | 1.000 |  |  |  |
| ETSI2 | <— | ETSI | .964 | .055 | 17.547 | *** |
| ETSI3 | <— | ETSI | 1.113 | .060 | 18.440 | *** |
| ERSI1 | <— | ERSI | 1.000 |  |  |  |
| ERSI2 | <— | ERSI | 1.301 | .082 | 15.862 | *** |
| ERSI3 | <— | ERSI | 1.024 | .064 | 16.005 | *** |
| ERSI4 | <— | ERSI | 1.073 | .076 | 14.114 | *** |
| ETSI4 | <— | ETSI | 1.025 | .063 | 16.324 | *** |

IFI = 0.977, TLI = 0.966, CFI = 0.977, all close to 1; RMSEA = 0.09, close to 0; the path coefficients are statistically significant at the level of P < 0.001(see Table 5). This factor structure has passed the validation. This study is effective in dividing and measuring all the variables.

## 4.2 Assessing the hypotheses

This paper performs structural equation modeling (SEM) to test the research model and the hypotheses. The initial structural equation model is analyzed and calculated by AMOS software. The fitting results were CMIN/DF = 2.626, GFI = 0.861, NFI = 0.911, RFI = 0.889, IFI = 0.943, TLI = 0.929, CFI = 0.942, RMSEA = 0.088. On Table 6, it can be seen hypothesis 2 is that customer intelligence has a significant positive impact on exploratory service innovation is confirmed; hypothesis 3 and 4 are confirmed, indicating that opponent CI has a significant positive impact on exploratory and exploitative service innovation; hypothesis 5 and 6 are also confirmed that supplier CI has a significant positive impact on exploratory and exploitative service innovation. However, hypothesis 1 that customer CI has positive effect on exploitative service innovation is not confirmed. According to the opinions of many respondents, service businesses are more likely to use customer CI to develop new services to meet their needs. In exploitative innovations, they are inclined to adapt supplier and opponent CI in order to avoid innovation failure. Thus, compared with exploratory service innovation, the positive impact of customer intelligence on exploitative service innovation is not significant.

# 5. Discussion and implication

## 5.1 Conclusions

In the process of innovation in service businesses, CI plays an important role. Through empirical research, this paper proves that three dimensions of enterprise CI have different influences on service innovation. Customer CI has stronger effect on exploratory service innovation than on exploitative service innovation. Opponent CI and supplier CI have obvious positive effects on both exploratory and exploitative service innovation.

**Table 6. Non-standardized regression coefficient.**

|  |  |  | Estimate | S.E. | C.R. | P |
|---|---|---|---|---|---|---|
| ERSI | <— | CCI | .247 | .062 | 3.984 | *** |
| ERSI | <— | OCI | .163 | .048 | 3.416 | *** |
| ERSI | <— | SCI | .298 | .071 | 4.174 | *** |
| ETSI | <— | CCI | .150 | .056 | 2.670 | .008 |
| ETSI | <— | OCI | .265 | .045 | 5.908 | *** |
| ETSI | <— | SCI | .262 | .065 | 4.023 | *** |

## 5.2 Theoretical contributions

This study is a pioneer to examine how different dimensions of CI generate direct effects on innovation in service firms. While many previous studies often link customer CI, supplier CI to service innovation [50–52] without taking opponent CI into consideration, this study has proposed effects of opponent CI on exploratory and exploitative service innovation. Our empirical testing has found the effect of opponent CI to be strongly supported, with the two paths significant in the hypothesized directions. As a result, this contributes to the development of a more comprehensive account of opponents' behavior.

A recent study by Mohan and his partners [53] has also highlighted the important role of suppliers in service innovation process. Indeed, the inclusion of supplier knowledge and technology in any theoretical model to predict innovations is strongly warranted. Besides, not unexpected, customer CI has a significant effect on exploratory service innovation. This finding is consistent with prior results in literature about customer knowledge and innovation [54–56]. But its positive effect on exploitative service innovation is not strong. This result would be a complementary for previous literature of customer and service innovation.

## 5.3 Practical contributions

Based on the empirical research results of tourism service industry with typical service characteristics, this paper provides the following management enlightenment for China's service industry in using CI to improve service innovation.

1. CI is an indispensable motive force and source of service innovation in an industry as a whole or in an individual enterprise. Application of CI has a positive influence on both exploratory and exploitative service innovation.

2. For service enterprises, customer is God, who is the purchaser of products and services. All information about customers' consumption demand, satisfaction and other aspects is essential for the survival and development of new products and services. By analyzing customers' past purchasing and potential customer needs through interactive platforms, customer visits or market surveys, enterprises will make progress in service innovation. Customers' previous purchasing and potential demand can help enterprises develop new services and try new fields.

3. Opponent CI which has always been the focus of managers and scholars plays a key role in the survival, development and innovation of service enterprises. It is indispensable for enterprises to acquire and analyze opponent CI. Competitors' new products and services can become the object of imitation and inspiration source of exploitative service innovation. For service enterprises, imitating competitors has the advantages of less investment, higher efficiency and less risk. Therefore, many service enterprises usually start their own innovation activities by imitating competitors.

4. Suppliers are important partners. New technology and knowledge are both sources for service innovation. Some service industries, such as hotels, are mainly supplier-led in technological innovation [39]. Suppliers can provide enterprises with new service production and process control solutions, tools and equipment needed for new services. Therefore, it is necessary to select innovative suppliers and strengthen cooperation and exchanges with them.

## 5.4 Limitations and future research

Firstly, this study got data mainly from tourism enterprises, such as tourist attractions, hotels, food and crafts companies, travel agencies and so on. This study considers tourism enterprise

as typical service businesses; therefore, it does not consider the different features of sub-types of service enterprises. Secondly, this study is conducted in east China, which has a unique cultural and economic environment in China. The generalizability of our findings to China's service enterprises will need to be confirmed with additional studies in different places to take account of the differences in culture and economy. In the future, further researches about service innovations of different types of service enterprises in different places are need.

## Supporting information

**S1 Data. English and Chinese quesitonnarie.**
(DOCX)

**S2 Data. Ethical statement.**
(DOCX)

**S3 Data. CI&SI(confimatory and amos-212).**
(SAV)

**S4 Data. CI&SI(exploratory-121).**
(SAV)

## Acknowledgments

We thank Cheng Changchun and Hua Hefeng for valuable research assistance.

## Author Contributions

**Conceptualization:** Yanli Bao.

**Data curation:** Yanli Bao.

**Formal analysis:** Yanli Bao.

**Methodology:** Yanli Bao.

**Project administration:** Yanli Bao.

**Resources:** Yanli Bao.

**Writing – original draft:** Yanli Bao.

**Writing – review & editing:** Yanli Bao.

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
