## [Decision Letter · Decision Letter 0]

19 Mar 2020

PONE-D-20-03099

Competitive Intelligence and Its impact on Innovations in Tourism Industry of China: an Empirical Research

PLOS ONE

Dear Dr. Bao,

Thank you for submitting your manuscript to PLOS ONE. After careful consideration, we feel that it has merit but does not fully meet PLOS ONE’s publication criteria as it currently stands. Therefore, we invite you to submit a revised version of the manuscript that addresses the points raised during the review process.

We would appreciate receiving your revised manuscript by May 03 2020 11:59PM. To enhance the reproducibility of your results, we recommend that if applicable you deposit your laboratory protocols in protocols.io, where a protocol can be assigned its own identifier (DOI) such that it can be cited independently in the future. For instructions see: http://journals.plos.org/plosone/s/submission-guidelines#loc-laboratory-protocols

We look forward to receiving your revised manuscript.

Kind regards,

Bing Xue, Ph.D.

Academic Editor

PLOS ONE

Journal Requirements:

2. Please provide additional details regarding participant consent. In the ethics statement in the Methods and online submission information, please ensure that you have specified (a) whether consent was informed and (b) what type you obtained (for instance, written or verbal, and if verbal, how it was documented and witnessed). If your study included minors, state whether you obtained consent from parents or guardians. If the need for consent was waived by the ethics committee, please include this information.

4. We note you have included a table to which you do not refer in the text of your manuscript. Please ensure that you refer to Table 6 and 7 in your text; if accepted, production will need this reference to link the reader to the Table.

Reviewers' comments:

Reviewer's Responses to Questions

**Comments to the Author**

1. Is the manuscript technically sound, and do the data support the conclusions?

Reviewer #1: Partly

Reviewer #2: No

Reviewer #3: Yes

2. Has the statistical analysis been performed appropriately and rigorously? 

Reviewer #1: I Don't Know

Reviewer #2: No

Reviewer #3: Yes

3. Have the authors made all data underlying the findings in their manuscript fully available?

Reviewer #1: No

Reviewer #2: No

Reviewer #3: Yes

4. Is the manuscript presented in an intelligible fashion and written in standard English?

Reviewer #1: No

Reviewer #2: Yes

Reviewer #3: Yes

5. Review Comments to the Author

Reviewer #1: The language needs to be modification. it is difficult to understand the author's meaning now.

The introduction of Literature review on CI is incomprehensive. What is the author's contribution?

The format of references cited in the paper is inconsistent, such as the second page Zhao chen2014.

Citation: Most of the literature is older and lacks of the latest research tracking.

Theoretical background: The description is lacking of logic. What is the author's intention to write this part?

Why are the hypothesis settings the same?How to distinguish in the questionnaire needs to be explained.

Method section: How are the 20 samples selected for the survey? Why is the survey designed at 7 latitudes? Who is the target of the survey? Data processing needs to be described in more detail.

The structural equation model mentioned in the abstract requires detailed to explain the research design of the method in the method section.

There is too little analysis of the results, the conclusions of the study design are not thoroughly analyzed, and there is little analysis related to the CI theme.

The discussion part is more like a conclusion. The discussion needs to be re-associated with the current research frontier and the research results obtained in this paper for a targeted discussion.

Reviewer #2: The facts given in the manuscript are already well known. I am unable to figure out any novel contribution. Secondly manuscript is organized poorly. For instance, 'Keywords' is just one word not two. After starting Section 3, authors immediately started subsection 3.1 without any content in Section 3.

Reviewer #3: 1. All discussion is too general without any comparison. I suggest authors should compare your results with others’ reports so as to find different conclusion.；

2. please have the paper thoroughly edited to address some grammatical errors that exist throughout the paper；

3. The authors state this approach is first used in this research field, How? There is no evidence in this paragraph

4. Where is the discussion of this literature? This is important.

6. PLOS authors have the option to publish the peer review history of their article (what does this mean?). If published, this will include your full peer review and any attached files.

Reviewer #1: No

Reviewer #2: No

Reviewer #3: No

---

## [Author Response · Author response to Decision Letter 0]

28 May 2020

Response to editor’s Comments

First of all, thank you very much for your very encouraging and inspiring feedback on my work and for your very constructive and helpful comments that have definitely improved the paper a great deal. I had studied all of your comments very carefully and tried to incorporate all of them into the current version of the paper. Please find more detailed descriptions of how I did that below. I have modified my paper to accommodate your concerns (major changes in the article are written as notes in the margin on each page for your quick perusal). In addition, I provided a point-by-point response to each of your comment. For ease of distinction, your originals comments are listed below in regular font, and my responses to those comments are shown in italics font.

Thank you very much for your information. I have edited my manuscript following PLOS ONE's style requirements.

2. Please provide additional details regarding participant consent. In the ethics statement in the Methods and online submission information, please ensure that you have specified (a) whether consent was informed and (b) what type you obtained (for instance, written or verbal, and if verbal, how it was documented and witnessed). If your study included minors, state whether you obtained consent from parents or guardians. If the need for consent was waived by the ethics committee, please include this information.

Thanks deeply for your suggestion. In section 3.1(Data collections), I have specified the participant consents. You can find:

 “Second, with the refined questionnaire, the investigator gets approval from the administrators of tourism companies and sends an invitation letter out through e-mail to express the need for collection of empirical data concerning service innovation experience in using CI. The administrators then forwarded the message to their staff via email and asked the receivers to click a hyperlink and redirected them to an online questionnaire system. Consequently, 400 invitation letters were sent to the staff in tourism industry through e-mail. In order to improve the return rate, another follow-up invitation letter was sent to non-responding staff with the same aforementioned procedure after a week. Finally, 362 staff had finished and returned the questionnaire. 

Following your suggestion, I added detailed description of methodology to section 3 Methodology and a questionnaire in both Chinese and English as supporting information. 

4. We note you have included a table to which you do not refer in the text of your manuscript. Please ensure that you refer to Table 6 and 7 in your text; if accepted, production will need this reference to link the reader to the Table.

Thank you for your insightful comment. I checked the tables and re-order them. Now I can ensure that all the tables are referred to in the manuscript.

Response to Reviewer 1’s Comments

First of all, thank you very much for your very encouraging and inspiring feedback on my work and for your very constructive and helpful comments that have definitely improved the paper a great deal. I had studied all of your comments very carefully and tried to incorporate all of them into the current version of the paper. Please find more detailed descriptions of how I did that below. I have modified my paper to accommodate your concerns (major changes in the article are written as notes in the margin on each page for your quick perusal). In addition, I provided a point-by-point response to each of your comment. For ease of distinction, your originals comments are listed below in regular font, and my responses to those comments are shown in italics font.

Reviewer #1: The language needs to be modification. it is difficult to understand the author's meaning now.

Thanks deeply for your suggestion. I have refined the sections of literature review, methodology, data analysis and discussion. Furthermore, in this revision, grammatical and writing style errors in the original version have been refined and edited by my friend who is a native English speaker.

The introduction of Literature review on CI is incomprehensive. What is the author's contribution?

Thank you very much for your inspiring feedback. I rewrote this part and included the definition of CI, its importance and contents, which are the basis of research framework and hypotheses. I simplified the expressions so that this part is comprehensible. The relevant content is shown as follows.

 Society of Competitive Intelligence Professional (SCIP) is an authoritative body in CI. In 2003, it defined CI as the systematic and ethical collection, analysis and management of external information that can affect the planning, decision-making and business operation. CI has been listed as the fourth reason for the survival of enterprises after capital, technology and talent[ ]. Entering the era of knowledge-based economy, the degree of informationization in China is getting higher and higher. Enterprise CI has gradually become one of the decisive factors for the survival and development of enterprises. Generally speaking, CI includes information about competitors, customers, suppliers and related technologies[ ]. Beal(2000) regards enterprise's customer intelligence, supplier intelligence, opponent intelligence as the competitive environment in which the enterprise lives[ ].

The format of references cited in the paper is inconsistent, such as the second page Zhao chen2014.

Thanks deeply for your inspection. I fixed this typing mistake in this new version. And I consulted the editor for the format of references and changed it. You can check it in the references of revised manuscript.

Citation: Most of the literature is older and lacks of the latest research tracking.

 Thanks for your helpful feedback. I have updated and cited the recent relevant publications including Barão, de Vasconcelos, Rocha, & Pereira(2017) ,Thornhill(2019);Urbinati, A., Chiaroni, D., Chiesa, V., Frattini, F., 2018; Melton, H. L., & Hartline, M. D. (2013); Kratzer, J., Meissner, D., Roud, V.(2017);Bianchi et al.(2016); Berchicci(2013). All of them are listed in the reference section.

Theoretical background: The description is lacking of logic. What is the author's intention to write this part?

Thanks for your insightful comment. According to your comment, I rewrote the section 2(Theoretical background). In this section, I first introduced the literature about 2.1(competitive intelligence) and 2.2(service innovation), then analyzed existing research findings about the relationship between them(2.3 CI and Service Innovation) in order to find out the research gap and lay the groundwork for making hypotheses. I hope you will find these changes in this section as corresponding to your helpful comment.

Why are the hypothesis settings the same?How to distinguish in the questionnaire needs to be explained.

Thank you for your insightful questions. The hypotheses settings are the same because I want to keep in the same style to express to relationship between CI and service innovation so that the readers can understand the hypotheses easily. In the questionnaire, I explain each variable clearly and the questionnaire was discussed intensively within our research institute and pre-tested independently with 5 managers from service businesses which were not included in the sample. These 5 managers all have more than 10 years working experiences in star hotels, travel agencies, tourist attractions, or other service companies. Based on the discussions, the questionnaire was modified. All the variables in both Chinese and English are listed here for your reference:

Customer CI refers to the competitive intelligence about customers: 顾客情报是指与顾客有关的竞争情报。

CCI1 We collect basic information of customers, including their name, age, occupation/profession,et.al. 我们收集客户的基本信息，包括他们的姓名、年龄、职业等。

CCI2 We collect and analyze customer need/demand about our services.我们收集并分析客户对我们服务的需求。

CCI3 We collect and analyze customer satisfaction and customer complaints.我们收集和分析客户满意度和客户投诉情况。

CCI4 We invite customers to participate in service innovation.我们邀请客户参与服务创新。

Opponent CI refers to the competitive intelligence about opponents:竞争对手情报是指关于竞争对手的情报：

OCI1 We collect and analyze opponents’ daily operation information. 我们收集并分析对手的日常运营信息。

OCI2 We keep eyes on R&D progress of our opponents.我们关注对手的研发进展。

OCI3 We pay attention to marketing of the new services/products from our opponents. 我们关注对手推销新服务/产品的情报。

Supplier CI refers to the competitive intelligence about suppliers:供应商情报是指关于供应商的竞争情报：

SCI1 We collect and analyze information about inventory of suppliers’ service/products.我们收集和分析有关供应商服务/产品库存的信息。

SCI2 We collect and analyze information about R&D of new service/products of suppliers.我们收集和分析供应商新服务/产品的研发信息。

SCI3 We collect and analyze information about marketing of service/products supplied by our suppliers. 我们收集并分析供应商提供的服务/产品的市场信息。

Exploitative service innovation(ETSI) refers to a small-scale and gradual innovation activity with the intention to improve the existing status. 利用式服务创新是一种小幅度、渐进的创新活动，其意图是对服务现状进行改进。

ETSI1 We make efforts to improve the applicability of existing service/skills in many related business areas.我们努力提高已有的技术/技能在多个相关业务领域的适用性。

ETSI2 We often use existing service/skills to increase the functions and types of products/services. 我们经常利用已有的技术/技能来增加产品/服务的功能和种类。

ETSI3 We often improve existing service/skills to meet current needs.我们经常对已有的技术/技能进行改良，以适应当前需要。

ETSI4 We often refine our accumulated business experience and applies it to the current business. 我们经常对公司积累的业务经验进行提炼，并应用于当前业务中。

Exploratory Service innovation(ERSI) refer to a large-scale and radical innovation activity with the intention of finding new possibilities. 探索式创新是一种大幅度的、激进的创新活动，其意图是寻找新的可能性。

ERSI1 We frequently develop brand-new market segments without relevant marketing experience. 我们经常开拓全新的、尚无相关营销经验的细分市场。

ERSI2 We often adopt business strategies/tactics that have not been adopted by other companies in the same industry. 我们经常采用同行业其他公司没有采用过的经营战略/战术。

ERSI3 We frequently use immature and risky new services/skills. 我们经常运用尚不成熟、有一定风险的新技术/技能。

ERSI4 We frequently develop new and radical products/services. 我们经常开发全新的、根本性变革的产品/服务。

Method section: How are the 20 samples selected for the survey? Why is the survey designed at 7 latitudes? Who is the target of the survey? Data processing needs to be described in more detail.

Thank you very much for your inspiring questions and suggestions. According to your suggestions, I rewrote section 3(Methodology） . I added how each construct is measured by questions in the survey, how the questions are modified, the targets of the survey and how the survey is delivered in service firms. The data processing is also described in Data Analysis section. The relevant content is shown as follows:

3.Methodology

3.1. Data collections

To test the hypotheses and the model, they had to be converted into a questionnaire. Each construct is represented by a set of indicators which form the questions in the survey. All questions were measured on a positive-to-negative 7-point Likert scale. Questions on the CI and service innovation give a statement and ask for the level of agreement on the following scale: "Strongly agree - predominantly agree - rather agree - neutral - rather disagree - predominantly disagree - strongly disagree." The questionnaire was discussed intensively within our research institute and pre-tested independently with 5 managers from service businesses which were not included in the sample. These 5 managers all have more than 10 years working experiences in star hotels, travel agencies, tourist attractions, or other service companies. Based on the discussions, the questionnaire was modified.

The firms selected for this study are employees of star hotels and tourism companies of more than 20 staff in China, because tourism industry has the typical characteristics of service industry and a huge amount in China.

Data was collected in two stages. First, in pre-survey, 100 questionnaires were distributed and 94 valid questionnaires were returned. In pre-survey, Cronbach’s alpha coefficient and factor load of the scale were calculated by SPSS 23 software, and the item was deleted according to relevant standards. Second, with the refined questionnaire, the investigator gets approval from the administrators of tourism companies and sends an invitation letter out through e-mail to express the need for collection of empirical data concerning service innovation experience in using CI. The administrators then forwarded the message to their staff via email and instructed the receivers to click a hyperlink and redirected them to an online questionnaire system. Consequently, 400 invitation letters were sent to the staff in tourism industry through e-mail. In order to improve the return rate, another follow-up invitation letter was sent to non-responding staff with the same aforementioned procedure after a week. Finally, 362 staff had finished and returned the questionnaire. Altogether 333 valid questionnaires were obtained after deleting unqualified questionnaires, with an effective return rate of 83.25%.

3.2. Measures

 Measurement items were selected on the basis of a careful literature review. The results from pre-survey showed that there is no particular bias. A description of the constructs and indicators is presented in Appendix 1. 

The scale of enterprise CI is modified by the relevant scales used in empirical researches. The scale of customer CI is made by revising Zhang Hongqi and his partners’ scale (2013)[ ]. Four items were adapted to measure the extend of customer CI(CCI), including basic customer information, customer demand, customer satisfaction and customer participation in innovation. The opponent intelligence(OCI) scale is prepared in 3 aspects, i.e. competitor’s daily operation CI, R&D CI and marketing CI of new services. The scale of supplier CI(SCI) is developed on the basis of Gales, Mansour-Cole (1995)[ ] and interviews with service business owners. Three are 3 items describing supplier CI including supplier inventory, R&D, and marketing.

The scale of service innovation adopts the scale of exploitative innovation(ETSI) and exploratory innovation(ERSI) developed by Fu Xiao et al. (2012)[ ] to assess the extent to which a firm has engaged in innovation activities and has implemented service innovation activities to improve existing service–market positions with 8 items. 

4. Data Analysis

The data analysis of this study was conducted using structural equation modeling (SEM) technique and followed the two-step approach of for assessing the measurement and structural models respectively[ ]. SEM is a powerful statistical research technique and it is very flexible in the types of theoretical models to be tested for analyzing the causal relationships between multiple-item constructs[ ]. In addition, SPSS and AMOS are adopted as the tools of data analysis.

The structural equation model mentioned in the abstract requires detailed to explain the research design of the method in the method section.

Thank you for your suggestion. I added content about SEM in Section 4 (Data analysis).The relevant content is shown as follows: The data analysis of this study was conducted using structural equation modeling (SEM) technique and followed the two-step approach of for assessing the measurement and structural models respectively[ ]. SEM is a powerful statistical research technique and it is very flexible in the types of theoretical models to be tested for analyzing the causal relationships between multiple-item constructs[ ]. In addition, SPSS and AMOS are adopted as the tools of data analysis.

There is too little analysis of the results, the conclusions of the study design are not thoroughly analyzed, and there is little analysis related to the CI theme.

The discussion part is more like a conclusion. The discussion needs to be re-associated with the current research frontier and the research results obtained in this paper for a targeted discussion.

Thanks for your helpful comment and suggestion. I rewrote the relevant contents and associated my results with the current research findings. The relevant contents are as follows: 

5. Discussion And Implication

5.1 Conclusions

In the process of innovation in service businesses, CI plays an important role. Through empirical research, this paper proves that three dimensions of enterprise CI have different influences on service innovation. Customer CI has stronger effect on exploratory service innovation than on exploitative service innovation. Opponent CI and supplier CI have obvious positive effects on both exploratory and exploitative service innovation. 

5.2 Theoretical contributions

This study is a pioneer to examine how different dimensions of CI generate direct effects on innovation in service firms. While many previous studies often link customer CI, supplier CI to service innovation[ ],[ ],[ ] without taking opponent CI into consideration, this study has proposed effects of opponent CI on exploratory and exploitative service innovation. Our empirical testing has found the effect of opponent CI to be strongly supported, with the two paths significant in the hypothesized directions. As a result, this contributes to the development of a more comprehensive account of opponents’ behavior. 

A recent study by Mohan and his partners[ ]has also highlighted the important role of suppliers in service innovation process. Indeed, the inclusion of supplier knowledge and technology in any theoretical model to predict innovations is strongly warranted. Besides, not unexpected, customer CI has a significant effect on exploratory service innovation. This finding is consistent with prior results in literature about customer knowledge and innovation[ ],[ ],[ ]. But its positive effect on exploitative service innovation is not strong. This result would be a complementary for previous literature of customer and service innovation.

5.3 Practical contributions

Based on the empirical research results of tourism service industry with typical service characteristics, this paper provides the following management enlightenment for China’s service industry in using CI to improve service innovation.

(1) CI is an indispensable motive force and source of service innovation in an industry as a whole or in an individual enterprise. Application of CI has a positive influence on both exploratory and exploitative service innovation. 

(2) For service enterprises, customer is God, who is the purchaser of products and services. All information about customers’ consumption demand, satisfaction and other aspects is essential for the survival and development of new products and services. By analyzing customers’ past purchasing and potential customer needs through interactive platforms, customer visits or market surveys, enterprises will make progress in service innovation. Customers’ previous purchasing and potential demand can help enterprises develop new services and try new fields.

(3) Opponent CI which has always been the focus of managers and scholars plays a key role in the survival, development and innovation of service enterprises. It is indispensable for enterprises to acquire and analyze opponent CI. Competitors’ new products and services can become the object of imitation and inspiration source of exploitative service innovation. For service enterprises, imitating competitors has the advantages of less investment, higher efficiency and less risk. Therefore, many service enterprises usually start their own innovation activities by imitating competitors.

(4) Suppliers are important partners. New technology and knowledge are both sources for service innovation. Some service industries, such as hotels, are mainly supplier-led in technological innovation[ ]. Suppliers can provide enterprises with new service production and process control solutions, tools and equipment needed for new services. Therefore, it is necessary to select innovative suppliers and strengthen cooperation and exchanges with them. 

5.4 Limitations and future research

Firstly, this study got data mainly from tourism enterprises, such as tourist attractions, hotels, food and crafts companies, travel agencies and so on. This study considers tourism enterprise as typical service businesses, therefore, it does not consider the different features of sub-types of service enterprises. Secondly, this study is conducted in east China, which has a unique cultural and economic environment in China. The generalizability of our findings to China’s service enterprises will need to be confirmed with additional studies in different places to take account of the differences in culture and economy. In the future, further researches about service innovations of different types of service enterprises in different places are need. 

Response to Reviewer 2’s Comments

Reviewer #2: The facts given in the manuscript are already well known. I am unable to figure out any novel contribution. Secondly manuscript is organized poorly. For instance, 'Keywords' is just one word not two. After starting Section 3, authors immediately started subsection 3.1 without any content in Section 3.

First of all, thank you very much for your very encouraging and inspiring feedback on my work and for your very constructive and helpful comments that have definitely improved the paper a great deal. I had studied all of your comments very carefully and re-organized my manuscript. According to your comment, the mistake of “keywords” is corrected, and the novel contribution is added in section 5( Discussion And Implication) as the theoretical and practical contributions. In this revision, grammatical and writing style errors in the original version have been refined and edited by my friend who is a native English speaker.

The relevant contents are as follows: 

5.2 Theoretical contributions

This study is a pioneer to examine how different dimensions of CI generate direct effects on innovation in service firms. While many previous studies often link customer CI, supplier CI to service innovation[ ],[ ],[ ] without taking opponent CI into consideration, this study has proposed effects of opponent CI on exploratory and exploitative service innovation. Our empirical testing has found the effect of opponent CI to be strongly supported, with the two paths significant in the hypothesized directions. As a result, this contributes to the development of a more comprehensive account of opponents’ behavior. 

A recent study by Mohan and his partners[ ]has also highlighted the important role of suppliers in service innovation process. Indeed, the inclusion of supplier knowledge and technology in any theoretical model to predict innovations is strongly warranted. Besides, not unexpected, customer CI has a significant effect on exploratory service innovation. This finding is consistent with prior results in literature about customer knowledge and innovation[ ],[ ],[ ]. But its positive effect on exploitative service innovation is not strong. This result would be a complementary for previous literature of customer and service innovation.

Response to Reviewer 3’s Comments

First of all, thank you very much for your very encouraging and inspiring feedback on my work and for your very constructive and helpful comments that have definitely improved the paper a great deal. I had studied all of your comments very carefully and tried to incorporate all of them into the current version of the paper. Please find more detailed descriptions of how I did that below. I have modified our paper to accommodate your concerns (major changes in the article are written as notes in the margin on each page for your quick perusal). In addition, I provided a point-by-point response to each of your comment. For ease of distinction, your originals comments are listed below in regular font, and my responses to those comments are shown in italics font.

Reviewer #3: 1. All discussion is too general without any comparison. I suggest authors should compare your results with others’ reports so as to find different conclusion.；

Than you very much for you helpful suggestion. I rewrote this the conclusion section and compare my results with the recent findings. Here are the comparisons in the revised edition:

A recent study by Mohan and his partners[ ]has also highlighted the important role of suppliers in service innovation process. Indeed, the inclusion of supplier knowledge and technology in any theoretical model to predict innovations is strongly warranted. Besides, not unexpected, customer CI has a significant effect on exploratory service innovation. This finding is consistent with prior results in literature about customer knowledge and innovation[ ],[ ],[ ]. But its positive effect on exploitative service innovation is not strong. This result would be a complementary for previous literature of customer and service innovation.

2. please have the paper thoroughly edited to address some grammatical errors that exist throughout the paper；

Thanks deeply for your suggestion. We have refined the sections of literature review, research method, data analysis and discussion. Furthermore, in this revision, grammatical and writing style errors in the original version have been refined and edited by my friend who is a native English speaker.

3. The authors state this approach is first used in this research field, How? There is no evidence in this paragraph.

Thanks a lot for your helpful comment. I rewrote this part with some evidence in this part. The relevant contents are as follows:

This study is a pioneer to examine how different dimensions of CI generate direct effects on innovation in service firms. While many previous studies often link customer CI, supplier CI to service innovation[ ],[ ],[ ] without taking opponent CI into consideration, this study has proposed effects of opponent CI on exploratory and exploitative service innovation. Our empirical testing has found the effect of opponent CI to be strongly supported, with the two paths significant in the hypothesized directions. As a result, this contributes to the development of a more comprehensive account of opponents’ behavior. 

4. Where is the discussion of this literature? This is important. 

Thank you very much for your insightful suggestion. I rewrote the discussion section which now consists of 5.1 Conclusions, 5.2 Theoretical contributions, 5.3 Practical contributions, 5.4 Limitations and future research. The relevant contents are as follows:

5. Discussion And Implication

5.1 Conclusions

In the process of innovation in service businesses, CI plays an important role. Through empirical research, this paper proves that three dimensions of enterprise CI have different influences on service innovation. Customer CI has stronger effect on exploratory service innovation than on exploitative service innovation. Opponent CI and supplier CI have obvious positive effects on both exploratory and exploitative service innovation. 

5.2 Theoretical contributions

This study is a pioneer to examine how different dimensions of CI generate direct effects on innovation in service firms. While many previous studies often link customer CI, supplier CI to service innovation[ ],[ ],[ ] without taking opponent CI into consideration, this study has proposed effects of opponent CI on exploratory and exploitative service innovation. Our empirical testing has found the effect of opponent CI to be strongly supported, with the two paths significant in the hypothesized directions. As a result, this contributes to the development of a more comprehensive account of opponents’ behavior. 

A recent study by Mohan and his partners[ ]has also highlighted the important role of suppliers in service innovation process. Indeed, the inclusion of supplier knowledge and technology in any theoretical model to predict innovations is strongly warranted. Besides, not unexpected, customer CI has a significant effect on exploratory service innovation. This finding is consistent with prior results in literature about customer knowledge and innovation[ ],[ ],[ ]. But its positive effect on exploitative service innovation is not strong. This result would be a complementary for previous literature of customer and service innovation.

5.3 Practical contributions

Based on the empirical research results of tourism service industry with typical service characteristics, this paper provides the following management enlightenment for China’s service industry in using CI to improve service innovation.

(1) CI is an indispensable motive force and source of service innovation in an industry as a whole or in an individual enterprise. Application of CI has a positive influence on both exploratory and exploitative service innovation. 

(2) For service enterprises, customer is God, who is the purchaser of products and services. All information about customers’ consumption demand, satisfaction and other aspects is essential for the survival and development of new products and services. By analyzing customers’ past purchasing and potential customer needs through interactive platforms, customer visits or market surveys, enterprises will make progress in service innovation. Customers’ previous purchasing and potential demand can help enterprises develop new services and try new fields.

(3) Opponent CI which has always been the focus of managers and scholars plays a key role in the survival, development and innovation of service enterprises. It is indispensable for enterprises to acquire and analyze opponent CI. Competitors’ new products and services can become the object of imitation and inspiration source of exploitative service innovation. For service enterprises, imitating competitors has the advantages of less investment, higher efficiency and less risk. Therefore, many service enterprises usually start their own innovation activities by imitating competitors.

(4) Suppliers are important partners. New technology and knowledge are both sources for service innovation. Some service industries, such as hotels, are mainly supplier-led in technological innovation[ ]. Suppliers can provide enterprises with new service production and process control solutions, tools and equipment needed for new services. Therefore, it is necessary to select innovative suppliers and strengthen cooperation and exchanges with them. 

5.4 Limitations and future research

Firstly, this study got data mainly from tourism enterprises, such as tourist attractions, hotels, food and crafts companies, travel agencies and so on. This study considers tourism enterprise as typical service businesses, therefore, it does not consider the different features of sub-types of service enterprises. Secondly, this study is conducted in east China, which has a unique cultural and economic environment in China. The generalizability of our findings to China’s service enterprises will need to be confirmed with additional studies in different places to take account of the differences in culture and economy. In the future, further researches about service innovations of different types of service enterprises in different places are need.

---

## [Decision Letter · Decision Letter 1]

8 Jul 2020

Competitive Intelligence and Its impact on Innovations in Tourism Industry of China: An Empirical Research

PONE-D-20-03099R1

Dear Dr. Bao,

We’re pleased to inform you that your manuscript has been judged scientifically suitable for publication and will be formally accepted for publication once it meets all outstanding technical requirements.

Kind regards,

Bing Xue, Ph.D.

Academic Editor

PLOS ONE

Additional Editor Comments (optional):

Reviewers' comments:

Reviewer's Responses to Questions

**Comments to the Author**

1. If the authors have adequately addressed your comments raised in a previous round of review and you feel that this manuscript is now acceptable for publication, you may indicate that here to bypass the “Comments to the Author” section, enter your conflict of interest statement in the “Confidential to Editor” section, and submit your "Accept" recommendation.

Reviewer #2: All comments have been addressed

2. Is the manuscript technically sound, and do the data support the conclusions?

Reviewer #2: Yes

3. Has the statistical analysis been performed appropriately and rigorously? 

Reviewer #2: Yes

4. Have the authors made all data underlying the findings in their manuscript fully available?

Reviewer #2: Yes

5. Is the manuscript presented in an intelligible fashion and written in standard English?

Reviewer #2: Yes

6. Review Comments to the Author

Reviewer #2: The manuscript has been refined as per comments given. The authors show that opponent CI

and supplier CI have positive influence on both exploratory and exploitative service innovation. This is fine, I recommend that the paper be accepted and take following measure before:

1. Can authors just address other methods, maybe statistical to tackle the problem (for the sake of future direction)

2. Limitations of research is very important to mention.

3. Again authors have started subsection immediately after section without any content in section, this has to be avoided.

4. Equations to be written properly using equation editor.

5. Improve the language of the manuscript.

7. PLOS authors have the option to publish the peer review history of their article (what does this mean?). If published, this will include your full peer review and any attached files.

Reviewer #2: No

---

## [Editor Report · Acceptance letter]

14 Jul 2020

PONE-D-20-03099R1 

Competitive Intelligence and Its impact on Innovations in Tourism Industry of China: An Empirical Research 

Dear Dr. Bao:

I'm pleased to inform you that your manuscript has been deemed suitable for publication in PLOS ONE. Congratulations! Your manuscript is now with our production department. 

Kind regards, 

on behalf of

Professor Bing Xue 

Academic Editor

PLOS ONE